# ALIGNMENT DOES MATTER: ENABLES PURE-SPEECH-TOKEN DIALOGUE WITH FROZEN TEXT LLMS

## ABSTRACT

End-to-end speech dialogue models typically inherit knowledge from pretrained text LLMs by continually learning to model speech tokens. Thus, alignment between speech tokens and text tokens becomes critical: effective alignment enables more efficient transfer of LLM knowledge from text to speech. A well-known challenge lies in the sequence length mismatch between speech tokens and text tokens. But is matching them on average (i.e., making one second of speech map to the same number of tokens as the corresponding text) truly optimal? The answer remains unknown. In this paper, we aim to discover the optimal speech sequence length—equivalently, the frame rate of speech tokens—for inheriting text LLM knowledge. First, we observe that when forcing speech length to approach text length under a standard LLM architecture (single VQ + single LM head), information loss becomes a bottleneck. To address this, we propose a new method that enables scaling the VQ codebook capacity up to nearly 300 bits per frame, coupled with an efficient audio LM head. This design preserves sufficient speech information under aggressive downsampling while aligning sequence lengths more closely with text tokens, all without modifying the base text LLM backbone. Next, we explore three alignment tasks—speech→text, text→speech, and speech→speech—under different speech token frame rates, while keeping the text LLM parameters frozen. Furthermore, we find that speech and text representation still occupy distinct latent subspaces in the LLM. To mitigate this gap, we introduce a representation alignment objective to further strengthen cross-modal alignment. Experiments show that with only alignment, a frozen text LLM can already perform pure speech-to-speech QA, achieving comparable results on speech QA benchmarks.

## 1 INTRODUCTION

Voice interaction is emerging as a core interface for AI assistants. They promise human-machine interaction by directly processing speech as both input and output. Current spoken-dialogue systems can broadly be divided into two categories (Arora et al., 2025; Ji et al., 2024). The first is the cascaded paradigm, which pipelines ASR → text LLM → TTS. Such systems incur latency and often discard speech cues during the intermediate text stage. The other paradigm is the end-to-end spoken dialogue model (Défossez et al., 2024; Comanici et al., 2025; Hurst et al., 2024; Xu et al., 2025; Ding et al., 2025; Huang et al., 2025). These models reduce latency and preserve speech nuance by operating on speech tokens throughout. However, these methods rely on continuing to adapt text LLMs for speech token modeling, which raises concerns about knowledge retention and efficient transfer across modalities.

This motivates a central question for end-to-end speech-dialogue models: how can the text LLM knowledge be effectively transferred into the speech modality? Existing systems often proceed by continuing the training of a pretrained text LLM using large volumes of speech data and paired speech–text corpora. While this approach can yield strong empirical performance, the collection and alignment of such data is costly, and the required computation is substantial. More critically, fine-tuning the text LLM in this way can lead to catastrophic forgetting, whereby the model's original text capabilities degrade. The goal of this continued training is to equip the model with the ability to both understand and generate newly introduced speech tokens. But a fundamental question remains unanswered: what kind of speech tokenization is best suited for speech dialogue so as to enable

efficient transfer of text-based reasoning and knowledge into the speech domain? In other words, the space of possible speech token frame rates and speech representations that optimally bridge text-to-speech modeling is still largely unexplored.

In this paper, we investigate what kind of speech tokenization can be aligned with text tokens so as to effectively inherit the capabilities of pretrained text LLMs. To isolate this factor, we freeze the text LLM throughout our study; by keeping textual representations fixed, we can more clearly examine which forms of speech tokenization best align with text. We study this problem from two perspectives. First, from the perspective of frame rate, we compare against the average text token rate of 3.32 Hz, and systematically test speech frame rates ranging from 50 Hz, 25 Hz, 12.5 Hz, 6.25 Hz, 4.17 Hz, 3.33 Hz, and 3.125 Hz down to 2.08 Hz—thus covering regimes where speech tokens are longer than, comparable to, or shorter than text tokens. Designing such short speech tokenizations has not been attempted before, as severe information bottlenecks naturally emerge. To address this issue, we propose a new framework that both mitigates information loss at very low frame rates and enables efficient multi-token prediction. Second, from the perspective of representation alignment, we further introduce an alignment objective that explicitly encourages speech and text embeddings to lie in a shared latent space, thereby closing the semantic gap between the two modalities.

Our contributions are threefold:

1. **Length alignment and new architecture.** We systematically investigate speech token frame rates ranging from 50 Hz down to 2.08 Hz, covering regimes longer than, comparable to, and shorter than text tokens. To overcome the severe information bottleneck at low frame rates, we propose a new architecture that enables scalable VQ codebook size and an efficient audio LM head.

2. **Representation alignment.** We introduce a representation alignment objective that explicitly encourages speech and text embeddings to reside in a shared latent space, mitigating the semantic gap.

3. **Empirical insights for further speech token design.** Our experiments demonstrate that pure speech-token dialogue can be realized with a frozen text LLM. Beyond empirical results, our findings also provide guidance for future speech token design, offering practical insights into how tokenization choices affect the transfer of text LLM knowledge into the speech modality.

## 2 RELATED WORK

### 2.1 SPOKEN DIALOGUE MODELING

Recent research on speech LLMs can be broadly divided into two architectural families: cascaded pipelines and end-to-end models. Cascaded systems decompose the problem into three separate modules, offering greater flexibility since each stage can leverage off-the-shelf expert models. However, such systems may discard paralinguistic information—such as emotion or speaker traits—during the early ASR stage. Moreover, they are prone to severe error propagation and high interaction latency, which hinders real-time application.

End-to-end systems aim to align speech representations directly with the hidden space of LLMs, enabling them to understand and generate speech seamlessly. For instance, SpeechGPT (Zhang et al., 2023) introduces a progressive alignment methodology that integrates existing LLMs with quantized self-supervised representations to enable spoken dialogue. LLaMA-Omni (Fang et al., 2024) employs a streaming decoder along with a CTC-based strategy for simultaneous speech generation. Interleaved methods like Spirit-LM (Nguyen et al., 2025) and GLM4Voice (Zeng et al., 2024) can smoothly alternate between text and speech tokens, significantly improving cross-modal alignment. Nevertheless, these approaches demand large amounts of interleaved text-speech corpus and involve complex data collection pipelines. Parallel paradigms, such as Mini-Omni (Chen et al., 2024) and SLAM-Omni (Xie & Wu, 2024), enhance alignment training efficiency by modeling text and speech tokens in parallel through autoregressive training. However, they still require explicit text tokens during inference, which undermines the simplicity of end-to-end modeling. In contrast, our method features efficient alignment training, while eliminating unnecessary text tokens in the output side.

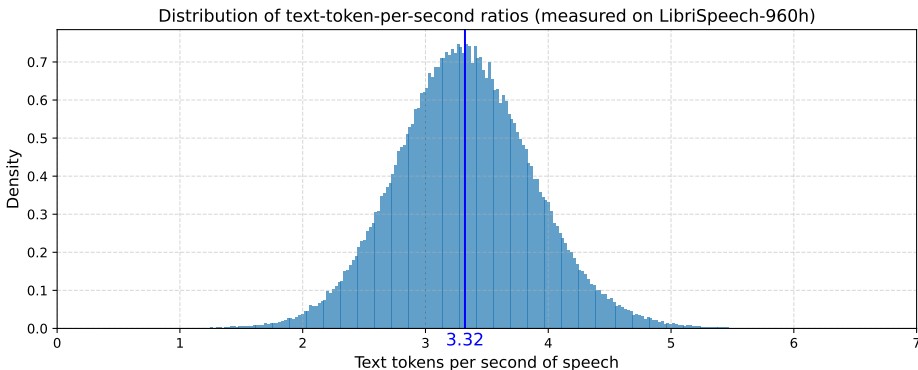

Figure 1: Distribution of text-token-per-second ratios on LibriSpeech-960h (Panayotov et al., 2015), where text transcriptions are tokenized with the Qwen3 tokenizer. The blue vertical line indicates the mean value.

## 2.2 SPEECH TOKENIZER

Speech tokenizers, which discretize continuous audio signals to accommodate the nature of language models, play a crucial role in speech language modeling. They can be broadly classified into acoustic tokenizers and semantic tokenizers. Acoustic tokenizers, largely derived from neural audio codecs (Défossez et al., 2022; Kumar et al., 2023), are designed to reconstruct high-fidelity waveforms. Owing to their superior ability to preserve acoustic details, they are widely used in speech generation (Wang et al., 2023) and editing (Peng et al., 2024) systems. In contrast, semantic tokenizers place greater emphasis on capturing linguistic content, which facilitates alignment with existing LLMs. These tokens are typically obtained by quantizing intermediate features from self-supervised learning models or ASR systems. For example, SpeechGPT (Zhang et al., 2023) and LLaMA-Omni (Fang et al., 2025) utilize discretized HuBERT (Hsu et al., 2021) tokens; SLAM-Omni (Chen et al., 2024) and GLM4Voice (Zeng et al., 2024) apply single-layer quantization to intermediate features of Whisper; while Moshi (Défossez et al., 2024) adopts a residual vector quantization (Lee et al., 2022) scheme and distills semantic information into the first layer. Although these methods can be integrated into LLMs, there remains a lack of systematic analysis regarding key design choices of semantic tokenizers—such as how token rate and quantization methods affect cross-modal alignment performance—in the context of spoken dialogue modeling.

## 3 LENGTH ALIGNMENT

### 3.1 PRELIMINARY

We begin by examining the length ratio between speech and text tokens. Using the LibriSpeech-960h corpus (about 256k speech-text pairs), we compute the ratio of text token length to speech duration, where transcriptions with preserved capitalization and punctuation are taken from LibriSpeech-PC (Meister et al., 2023) to retain semantic content. Text is tokenized with the Qwen3 tokenizer (Yang et al., 2025). As shown in Fig. 1, the average ratio is approximately 3.32 tokens per second of speech.

This raises a natural question: if we design a speech tokenizer whose frame rate closely matches the text token rate, would this lead to more effective alignment?

However, Most prior speech LLM architectures and speech tokenizers (Zeng et al., 2024; Zhang et al., 2023; Nguyen et al., 2025; Hassid et al., 2023; Lakhotia et al., 2021) use a standard design: a single Transformer backbone that embeds single-stream speech tokens and predicts them with a single LM head. But this paradigm suffers from a critical limitation: an *information bottleneck* emerges once the frame rate of speech tokens is reduced. To verify and better understand this phenomenon, we conduct a preliminary experiment where the encoder of Whisper-Large-v3 is used to extract speech features (Zeng et al., 2024; Liu et al., 2025; Ma et al., 2025; Fang et al., 2024). These

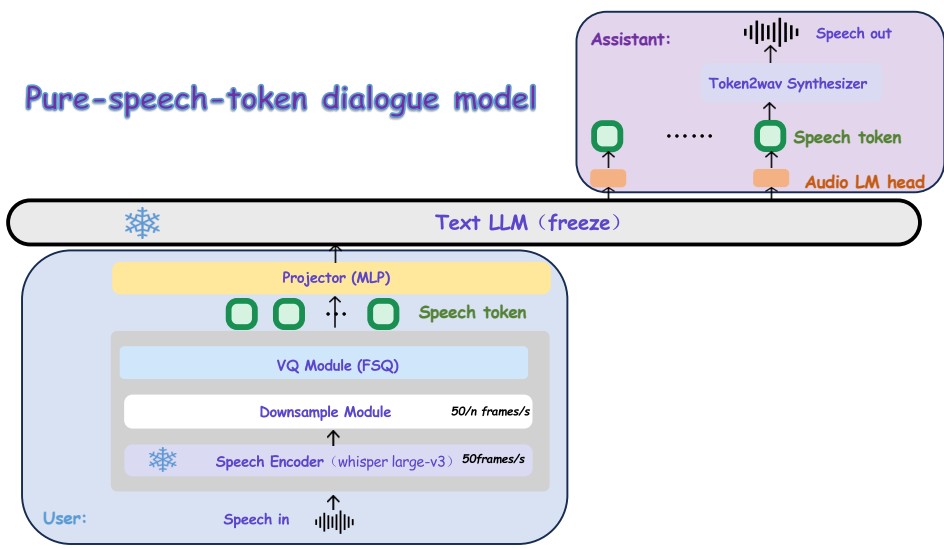

Figure 2: Architecture of our pure-speech-token dialogue model

features are then passed through a downsampling module to reach the target temporal resolution, followed by vector quantization (VQ) and a projector that maps the quantized embeddings into the LLM embedding space, which the freeze LLM serves as the ASR decoder. The pseudocode is provided in Appendix 1. In this preliminary setup, we start with an ASR supervised tokenizer both to follow existing designs (Zeng et al., 2024; Du et al., 2024) and also because our primary aim is to analyze speech–text alignment.

As illustrated in Fig. 3, with a codebook size of 4k ($2^{12}$), commonly used in prior work—performance degrades sharply once the temporal resolution falls below a downsampling rate of 4. We further scale the codebook size up to 262k($2^{18}$), even larger than the vocabularies of modern text LLMs (e.g., LLaMA-3 (Dubey et al., 2024): 128k, Qwen-3(Yang et al., 2025): 152k, GPT-OSS(Agarwal et al., 2025): 201k). Nevertheless, we find that performance still drops sharply once the temporal resolution falls below a downsampling rate of 8 (corresponding to 6.25 Hz). As shown in Fig. 3, this degradation is directly tied to information loss caused by aggressive downsampling, which fundamentally limits the exploration of optimal speech frame rates, whether equal to the average text token rate of 3.32 Hz.

### 3.2 PROPOSED SOLUTION

To overcome these limitations, the most straightforward idea is to continue increasing the codebook size. For speech input, this is unproblematic — unlike in text, where one must learn a very large embedding metric, in the speech setting, we only quantize features and then project them into the text embedding space.

However, this becomes problematic during token prediction: increasing the vocabulary size directly increases the number of parameters in the LM head, and every time we predict the next token, the softmax must compute scores over the full vocabulary. This causes training and inference to slow down significantly. Furthermore, memory usage rises: model weights for a huge vocabulary become a major bottleneck. For example, if the hidden size is 4,096 and the vocabulary size is 200k, that's already 800M parameters; increasing the vocabulary size by ten times would mean 8B parameters just for the LM head, which is unacceptable.

Therefore, if we can solve the problem of predicting from a large codebook, the issue is largely resolved. Unlike text vocabularies, where each token typically corresponds to a subword or word, our speech codebook comes from FSQ quantization of continuous features. Let the feature dimension be $d$, and each scalar dimension has $L$ quantization levels in FSQ, which yields an implicit codebook of size $L^d$. In other words, this means that instead of treating the entire $L^d$ codebook as atomic, we

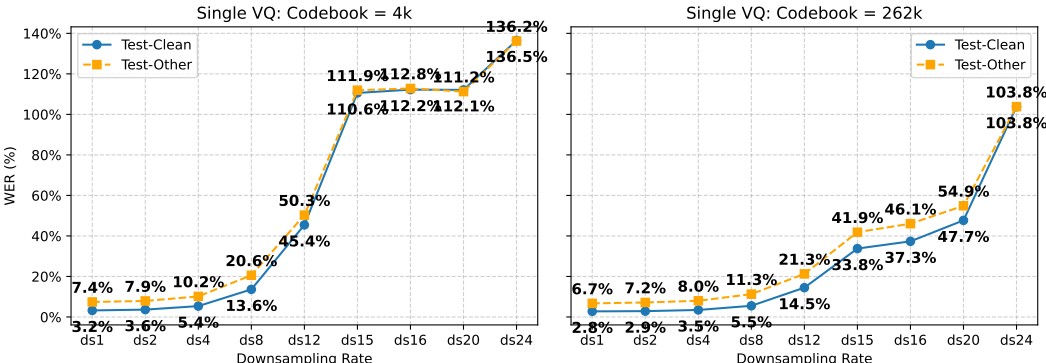

Figure 3: ASR WER versus the downsampling rate (ds) for two codebook sizes (4k vs. 256k) using librispeech test-clean and test-other. Here, ds1 (downsampling 1x) corresponds to 50Hz, ds2 to 25Hz, ds4 to 12.5Hz, ds8 to 6.25Hz,ds15 to 3.33Hz, ds16 to 3.125Hz, ds20 to 2.5Hz, and ds24 to 2.08Hz.

could predict each dimension's quantization level in parallel, or split the codebook into $n$ groups, each representing about $L^{d/n}$ combinations, and predict group-wise.

By giving each dimension (or each group) a learnable embedding tag, we could reuse the final classification head, thereby decomposing the large codebook prediction into smaller, factorized predictions. This would reduce the LM head parameter and computation cost dramatically while preserving expressive power. To push this further, we replace the usual linear head with a lightweight non-autoregressive transformer that predicts all groups' codes in parallel. This modification is to increase the expressive capacity of the head while keeping it efficient. Implementation details can be found in the Appendix 2

## 4 REPRESENTATION ALIGNMENT

After exploring sequence length alignment, there still remains a semantic gap: speech and text tokens may reside in distinct latent subspaces of the LLM. To address this, we introduce a representation alignment objective that explicitly encourages semantically corresponding speech and text embeddings to lie closer in the same hidden space.

A key design choice is that we operate on intermediate hidden states of a frozen LLM, rather than only on the embedding space. This layer-wise alignment allows semantic correspondence to be injected throughout the LLM hierarchy while keeping the backbone parameters fixed. In practice, for each speech–text pair, we extract hidden states from both modalities at selected LLM layers, average over the temporal dimension, and obtain normalized vectors $\widehat{h}_s, \widehat{h}_t \in \mathbb{R}^d$. We then optimize a contrastive objective of the InfoNCE form:

$$\mathcal{L}_{\text{align}} = -\sum_{i=1}^{B} \log \frac{\exp\big(\langle \widehat{h}_{s,i}, \widehat{h}_{t,i} \rangle / \tau\big)}{\sum_{j=1}^{B} \exp\big(\langle \widehat{h}_{s,i}, \widehat{h}_{t,j} \rangle / \tau\big)}, \tag{1}$$

where $B$ is the batch size, $\tau$ is a temperature hyperparameter, and $\langle \cdot, \cdot \rangle$ denotes the inner product (cosine similarity after normalization).

The overall training loss combines task supervision and alignment:

$$\mathcal{L} = \mathcal{L}_{\text{task}} + \lambda_{\text{align}} \, \mathcal{L}_{\text{align}}, \tag{2}$$

where $\lambda_{\text{align}}$ balances the contribution of the alignment term. This simple yet effective mechanism closes the modality gap without updating the frozen LLM, and ensures that speech and text tokens are embedded into a shared semantic space across hidden layers.

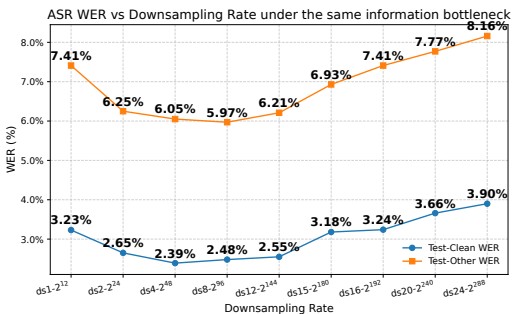
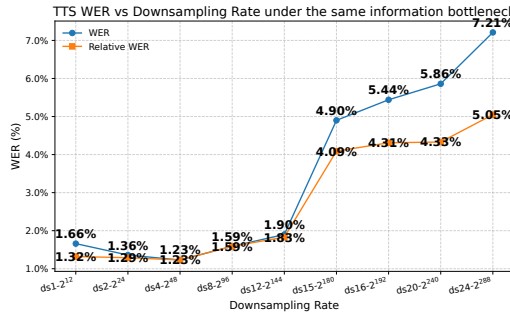

Figure 4: ASR WER on *LibriSpeech test-clean* and *test-other* at different downsampling rates (ds) under the same information bottleneck.

Figure 5: WER of TTS at different downsampling rates (ds) under the same information bottleneck.

## 5 EXPERIMENTS

We organize our experiments into two major parts: (i) length alignment, where we investigate the impact of different speech token frame rates on downstream performance; and (ii) representation alignment, where we assess how explicitly aligning hidden spaces improves semantic transfer across modalities. We use Qwen3-4B as the text LLM backbone model during the whole experiment.

### 5.1 LENGTH ALIGNMENT EXPERIMENTS

We structure our length alignment experiments into three stages. **Stage 1: Speech-to-text.** We first examine how many speech tokens per second are required to faithfully map to text tokens under a fixed information bottleneck. This stage evaluates how well a pretrained text LLM can process speech inputs at different frame rates when performing recognition. The speech tokens obtained here are also reused in subsequent experiments. **Stage 2: Text-to-speech.** Conditioned on the text tokens, we generate speech tokens using the frozen text LLM together with the newly introduced audio LM head. This experiment evaluates how well a pretrained LLM can produce speech tokens under different frame rates, thereby indicating which temporal resolution is most compatible with the text llm backbone. **Stage 3: Speech-to-speech QA.** Finally, we evaluate speech-only question answering, asking how well speech tokens at different frame rates can inherit the reasoning ability of the frozen text LLM and transfer it to a pure speech QA setting.

### 5.2 SPEECH-TO-TEXT

We begin with the most fundamental Speech-to-text setting: speech recognition. All models are trained on the LibriSpeech 960h corpus for 5 epochs with a batch size of 8 per GPU and a learning rate of $3 \times 10^{-4}$. Notably, we use the LibriSpeech-PC (Meister et al., 2023) transcripts that preserve case and punctuation, rather than normalized text. This design choice is motivated by our ultimate goal of transferring text capabilities into the speech domain; our experiments indicate that text normalization, while standard in ASR, degrades the transferability of text knowledge in speech-to-speech QA.

For evaluation, we report word error rate (WER) on *LibriSpeech test-clean* and *test-other*, where the references and the prediction are normalized using the Whisper normalizer[1].

Our experiments start from a baseline configuration of 50 Hz frame rate with a per-frame codebook size of $2^{12} = 4096$, corresponding to an information rate of $12 \times 50 = 600$ bits per second. To control for information throughput, we fix this bitrate across different downsampling factors. For example, when downsampling by $2\times$ (25 Hz), we increase the codebook size to $2^{24}$ (24 bits per

---

[1] https://github.com/openai/whisper/blob/main/notebooks/LibriSpeech.ipynb

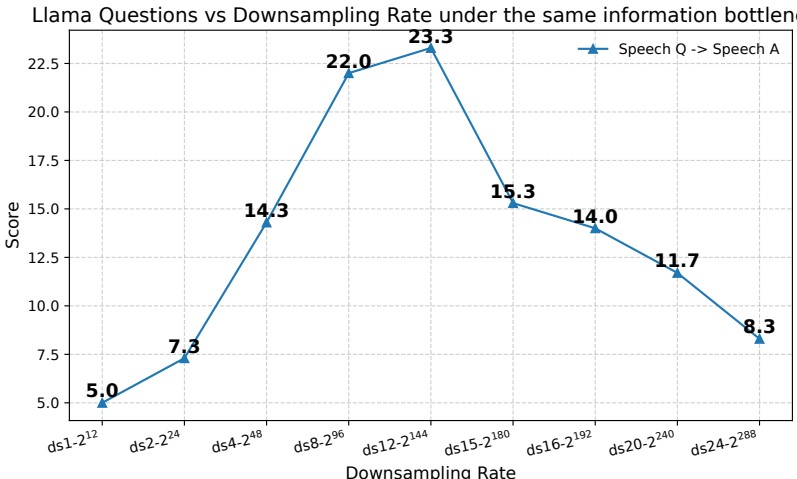

Figure 6: Speech QA results across different downsampling rates under the same information bottleneck.

frame), maintaining the same 600 bps. This scaling continues until $24\times$ downsampling (2.08 Hz), where each frame carries 288 bits, i.e., a codebook size of $2^{288}$.

The resulting performance trends are shown in Figure 4. Overall, ASR performance remains within a relatively narrow range (test-other: from 5.97 to 8.16, test-clean: from 2.39 to 3.90). Compared to the case of fixing the codebook size, this further validates our observation of the information bottleneck phenomenon.

In addition, we observe a U-shaped behavior with respect to downsampling: when the speech token sequence becomes shorter than the corresponding text length, WER increases; conversely, when the sequence is excessively long, performance also deteriorates, likely because the LLM backbone struggles with redundant temporal resolution. The most favorable region emerges at intermediate frame rates (12.5, 6.25, and 4.17Hz), yielding the lowest WER.

## 5.3 TEXT-TO-SPEECH

We next train the mapping from the text token to the speech token. For training, we reuse the speech tokens obtained from the ASR stage, which means freezing the downsampling module. Only the projector and the audio LM head are updated. We adopt exactly the same dataset and training hyperparameters as in the ASR experiments.

Since the speech token sequences are *reused* from the ASR stage, their text transcriptions are also generated by the previous stage's ASR model. This design avoids training a separate token-to-waveform synthesizer for each configuration, which would be computationally expensive and could introduce confounding variables. Since the ASR performance itself lies within a relatively narrow range across downsampling settings.

To further normalize differences in ASR quality, we additionally compute a *relative WER*, where the best-performing configuration is scaled to 1 and others are adjusted by their relative ASR capability. The results are shown in Figure 5. We observe a consistent trend: as downsampling increases, the WER monotonically worsens. More importantly, once the speech sequence length exceeds or matches the average text length, a sharp degradation occurs. This suggests that when speech tokens become much shorter than text tokens, the information density per token is too high, making it difficult for the frozen text LLM to accurately predict them.

Additionally, we tested the token2wav module of CosyVoice2 to convert speech tokens back into the speech waveform. We found that the synthesized audio content was accurate, despite a low frame rate, as our tokens contain sufficient information, as demonstrated by ASR results.

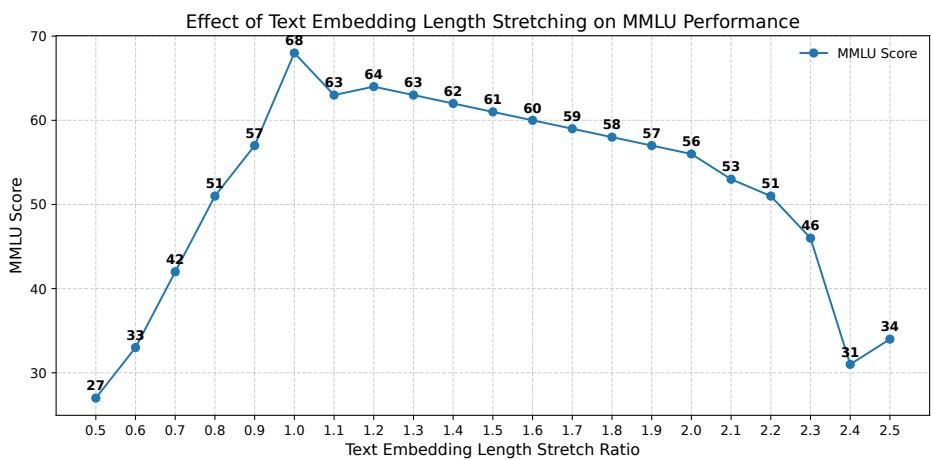

Figure 7: Effect of Text Embedding Length Stretching on MMLU Performance

## 5.4 SPEECH-TO-SPEECH

In this stage, we focus on speech QA. The speech tokens are reused from the ASR stage, and the model architecture is the same as in the TTS stage. In addition, the projector and audio head are initialized from the previous stage. For training, we adopt the InstructS2S-200k dataset (Fang et al., 2025), which contains 1,500 hours of speech QA data. During training, we include not only speech-to-speech QA pairs but also the corresponding speech-to-text QA and text-to-speech QA tasks as auxiliary tasks. The loss weight for speech-to-text QA is set to 5, while all other tasks are weighted as 1. We train the model for 3 epochs with a learning rate of $1 \times 10^{-4}$. As shown in Figure 6, the QA scores first increase and then decrease, with clear peaks at 4.17 Hz and 6.25 Hz.

## 5.5 ANALYSIS

We further conduct an auxiliary experiment on Qwen-4B using the MMLU benchmark to simulate the effect of stretching text embedding lengths. Specifically, we stretch the LLM's text embeddings by linear interpolation. While the alignment between speech and text is not strictly linear, it is monotonic, so this setup serves as a reasonable approximation. The results reveal that a pretrained text LLM is more tolerant to longer input lengths, whereas performance degrades sharply when the sequence is shortened. As shown in Figure 7, MMLU scores remain above 50 when the scaling factor lies between 0.8 and 2.2, suggesting this range is acceptable.

Combining this with Figure 6, we note that although the average text token rate is 3.32 Hz—closest to the 3.33 Hz (ds15) setting—this average masks a distribution: some utterances are shorter, others longer. The 3.33 Hz setting performs poorly because sequences shorter than $0.8\times$ the average degrade significantly. In contrast, settings such as 4.17 Hz ($\sim 1.25\times$ the text rate) and 6.25 Hz ($\sim 1.88\times$) fall within the acceptable 0.8–2.2 range and thus outperform the average-matching configuration. This also explains why speech QA achieves its best results at 4.17 Hz and 6.25 Hz rather than exactly matching the average text token rate.

## 5.6 REPRESENTATION ALIGNMENT EXPERIMENTS

We incorporate a simple contrastive alignment loss at all stages of training. For each speech clip, we extract the feature of its corresponding transcription through text LLM; We select a probe layer $\ell$ and average the hidden states over the temporal axis to obtain vectors $\widehat{h}_s, \widehat{h}_t \in \mathbb{R}^d$ (L2–normalized). We then optimize an InfoNCE objective with temperature $\tau$;

**Which LLM layer to align?** We investigate which LLM layer provides the most effective alignment signal by sweeping over four options: the text input embedding (EMB), an early layer ($\lfloor L/4 \rfloor$), a middle layer ($\lfloor L/2 \rfloor$), and a late layer ($\lfloor 3L/4 \rfloor$). We use the 4.17Hz frame rate setting as men-

|  | None | Emb | L/4 | L/2 | 3L/4 |
|---|---|---|---|---|---|
| Llama Questions | 23.3 | 21.7 | 25.0 | 30.7 | 27.7 |

Table 1: Effect of aligning at different LLM layers on Llama Questions. Alignment at the middle layer ($L/2$) provides the largest improvement.

tioned above. Experiments show that alignment at the mid layers yields the strongest performance, as summarized in Table 1.

## 5.7 COMPARE WITH OTHER METHODS

| Model | Trainable Params | Data | Web Q. | Llama Q. | Trivia QA |
|---|---|---|---|---|---|
| | | Moshi (Défossez et al., 2024) | | | |
| 7B | 7B | 7M hours | 9.2 | 21.0 | 7.3 |
| | | Scaling Speech-Text Pre-training with Synthetic Interleaved Data (Zeng et al., 2025) | | | |
| 9B | 9B | 600B Interleave | 15.9 | 50.7 | 26.5 |
| | 9B | No Interleaving | 0.1 | 2.3 | 0.2 |
| | 9B | 100B Interleave | 9.3 | 37.0 | 11.7 |
| | 9B | 200B Interleave | 13.3 | 44.0 | 18.7 |
| 1.5B | 1.5B | 600B Interleave | 5.4 | 18.3 | 4.6 |
| | 1.5B | No Interleaving | 0.0 | 1.3 | 0.0 |
| | | Ours | | | |
| 4B | $\sim$ 100M (Proj+Audio head) | 2.5k hours | 7.9 | 30.7 | 11.9 |
| 8B | $\sim$ 150M (Proj+Audio head) | 2.5k hours | 12.2 | 39.3 | 17.6 |

Table 2: Spoken Question Answering results (S2S only) on Web Questions, Llama Questions, and Trivia QA. Since our focus is on pure speech-token responses, we do not compare against models that first generate text tokens as guidance before producing speech tokens.

## 5.8 COMPARISON WITH OTHER METHODS

In this section, we compare our model against other spoken dialogue systems capable of producing pure speech-token responses. We further train a new model based on Qwen3-8B, with the same settings as the 4B version: a frame rate of 4.17 Hz and middle-layer alignment. Compared with Moshi, which leverages 700M hours of speech data with a 7B text backbone under full-parameter training, our 4B model—trained on only 2.5k hours of data with $\sim$100M trainable parameters—outperforms it across all three QA benchmarks. In addition, compared with methods that scale interleaved data (Zeng et al., 2025), our 8B model achieves results comparable to those requiring 100B–200B interleaved tokens and full 9B parameter training, despite using only 2.5k hours of speech data and updating $\sim$150M parameters. These results demonstrate the effectiveness of our alignment-based approach. We believe that the design of our speech token would bring insight to the future speech dialogue model.

## 5.9 CONCLUSION

We presented a simple recipe for enabling pure speech-token dialogue on top of a frozen text LLM. Our study highlights two alignment dimensions—length and representation—and shows that carefully chosen speech token rates, together with a scalable tokenizer and audio LM head design, can overcome the information bottleneck that arises under aggressive downsampling. A contrastive objective further closes the cross-modal gap without modifying the backbone. Empirically, the resulting system delivers competitive speech-to-speech QA with modest data and benefits from speech-text alignment.

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

# A APPENDIX

## A.1 PSEUDOCODE

Listing 1: Pseudocode of Preliminary ASR Experiment

```python
import whisper
from vector_quantize_pytorch import FSQ
from encodec.modules import SEANetEncoder

class Preliminary_ASR_Exp():
    def __init__(self, llm, model_args):
        # Frozen text LLM backbone (e.g., Qwen3-4B)
        # Used here as an ASR decoder
        self.llm = llm

        # Projector: map quantized speech features into the LLM
        #     embedding space
        self.projector = MLP(in_dim=model_args.vq_dim, out_dim=llm.
            hidden_size)

        # Frozen Whisper-Large-v3 encoder to extract speech
        #     representations
        self.pretrained_speech_encoder = whisper.load_model('large-v3'
            ) # frozen

        # FSQ quantizer: converts continuous features into discrete
        #     tokens
        self.vq = FSQ(
            levels = [8] * 4,          # codebook size = 4096
            # levels = [8] * 6,        # codebook size = 262k
        )

        # Downsampling module: reduces temporal resolution before
        #     quantization
        self.downsample_module = SEANetEncoder(
            channels=1280,
            dimension=model_args.vq_dim,
            n_filters=model_args.n_filters,
            ratios=[model_args.ds_rate]
        )

    def get_speech_embeddings(self, mel):
        # Step 1. Extract  Whisper features
        with torch.no_grad():
            speech_repr = self.pretrained_speech_encoder.embed_audio(
                mel)
        # Step 2. Downsample to target frame rate (controlled by
        #     ds_rate)
        z = self.downsample_module(speech_repr)
        # Step 3. Quantize into discrete tokens
        z_q, token = self.vq(z)
        # Step 4. Project into LLM embedding space
        return self.projector(z_q)

    def forward(self, Instruction_ids,  asr_transcription_labels,
            speech):
        # ASR Instruction: 'Repeat exactly what the user says word by
        #     word.'
        Instruction_text_emb = self.llm.get_input_embeddings()(
            Instruction_ids)

        # Speech embeddings at chosen frame rate
        speech_emb = self.get_speech_embeddings(speech)
```

```
49              # Feed both into frozen LLM to train ASR
50              out = self.llm(inputs_embeds=[Instruction_text_emb, speech_emb
                    ],
51                              labels=asr_transcription_labels)
52          return out
```

Listing 2: Pseudocode of our speech LLM Experiment

```python
1   import whisper
2   from vector_quantize_pytorch import FSQ
3   from encodec.modules import SEANetEncoder
4   from utils import MLP, NAR_Transformer,cross_entropy_loss
5
6   class Ours_SpeechLLM():
7       def __init__(self, llm, model_args):
8           # Frozen text LLM backbone (e.g., Qwen3-4B)
9           self.llm = llm
10
11          # Projector: map quantized speech features into the LLM
                embedding space
12          self.projector = MLP(in_dim=model_args.vq_dim, out_dim=llm.
                hidden_size)
13
14          # Frozen Whisper-Large-v3 encoder to extract speech
                representations
15          self.pretrained_speech_encoder = whisper.load_model('large-v3'
                ) # frozen
16
17          # FSQ quantizer: converts continuous features into discrete
                tokens
18          self.vq = FSQ(
19              levels = [8] * 4,           # group codebook size = 4096
20              num_codebooks = self.head_groups,       # Number of groups
21          )
22
23          # Lightweight Non-autoregressive Transformer head: predicts
                different groups quantized speech tokens in parallel
24          self.audio_head = NAR_Transformer(model_args.audio_head_cfg)
25
26          # Slot embeddings: learnable tags for each group
27          self.slot_embed = nn.Parameter(torch.randn(self.head_groups,
                self.llm.model.config.hidden_size))
28
29          # Downsampling module: reduces temporal resolution before
                quantization
30          self.downsample_module = SEANetEncoder(
31              channels=1280,
32              dimension=model_args.vq_dim,
33              n_filters=model_args.n_filters,
34              ratios=[model_args.ds_rate]
35          )
36
37      def get_speech_embeddings(self, mel):
38          # Step 1. Extract Whisper features
39          with torch.no_grad():
40              speech_repr = self.pretrained_speech_encoder.embed_audio(
                    mel)
41          # Step 2. Downsample to target frame rate
42          z = self.downsample_module(speech_repr)
43          # Step 3. Quantize into discrete tokens
44          z_q, token = self.vq(z)
45          # Step 4. Project into LLM hidden space
46          return self.projector(z_q), token
47
48      def predict_audio_tokens(self, hidden, target_tokens):
```

```
49        """
50
51        Use slot embeddings + audio head to predict different groups
52            quantized speech tokens in parallel
53        """
54        hidden_audio = hidden.unsqueeze(2).expand(-1, -1, self.
             head_groups, -1)
55        hidden_audio = hidden_audio + self.slot_embed.view(1, 1, self.
             head_groups, -1)
56        logits_audio = self.audio_head(hidden_audio)
57        loss = cross_entropy_loss(logits_audio, target_tokens)
58        return logits_audio, loss
59
60    def forward(self, text_input_ids, text_output_labels, speech_in,
             speech_out):
61        """
62        Forward pass covers three alignment experiments:
63        1) Speech-to-Text
64        2) Text-to-Speech
65        3) Speech-to-Speech
66        """
67
68        # ------------------------------
69        # 1. Speech -> Text
70        speech_in_emb, _ = self.get_speech_embeddings(speech_in)
71        out_st = self.llm(inputs_embeds=speech_in_emb,
72                          labels=text_output_labels)
73
74        # ------------------------------
75        # 2. Text -> Speech
76        speech_out_emb, speech_out_tokens = self.get_speech_embeddings
             (speech_out)
77        text_in_emb = self.llm.get_input_embeddings()(text_input_ids)
78        out_ts = self.llm(inputs_embeds=[text_in_emb, speech_out_emb],
79                          output_hidden_states=True)
80        hidden = out_ts['hidden_states'][-1]
81        loss_ts = self.predict_audio_tokens(hidden, speech_out_tokens)
82
83        # ------------------------------
84        # 3. Speech -> Speech
85        speech_in_emb, _ = self.get_speech_embeddings(speech_in)
86        speech_out_emb, speech_out_tokens = self.get_speech_embeddings
             (speech_out)
87        out_ss = self.llm(inputs_embeds=[speech_in_emb, speech_out_emb
             ],
88                          output_hidden_states=True)
89        hidden = out_ss['hidden_states'][-1]
90        loss_ss = self.predict_audio_tokens(hidden, speech_out_tokens)
91
92        return out_st, loss_ts, loss_ss
```

## B USE OF LARGE LANGUAGE MODELS

The large language model (LLM) was employed solely for language refinement, including improvements to grammar, spelling, clarity, and tone, on text originally crafted by the authors. The LLM did not introduce any substantive changes to the claims, data interpretation, or conclusions.

