# OpenReview forum: "Alignment Does Matter: Enables Pure-Speech-Token Dialogue with Frozen Text LLMs"
_ICLR.cc/2026/Conference — Submitted to ICLR 2026_

### Official Review · Reviewer_GATy · 2025-10-28

**Soundness:** 3
**Presentation:** 3
**Contribution:** 3
**Rating:** 4
**Confidence:** 4

**Summary:**

The main idea of this paper is to align the length of audio modality and the corresponding text modality for aligning audio modality with pre-trained text based LLMs.

The main contributions of the papers are:
(1) Identifying the information bottleneck when explicitly downsampling the audio frequency to match the text frequency. The key bottleneck lies within the codebook length when aggressively downsampling.
(2) Authors propose a multi-codebook approach where, the input representation are split into n groups and the predictions are performed in parallel for each n-groups using a non-autoregressive transformer.
(3) Additional InfoNCE based contrastive loss for aligning the speech representations and text representations (pooled over temporal axis).

Experimental results on Librispeech (960 hours) on ASR task validate the information bottleneck hypothesis with aggressive downsampling. Proposed approach of using multi-codebook alleviates the information bottleneck as evident from the ASR and TTS tasks.

**Strengths:**

1. The proposed information bottleneck experiment and its alleviation with multi-code book is interesting and novel.

2. Experimental results show the usefulness of the proposed approach in various settings such as speech-to-text, text-to-speech and speech-to-speech.

**Weaknesses:**

1. Core experiments use LibriSpeech and InstructS2S. These are clean, curated, and mainly limited to English. Claims about cross-modal alignment and generalizability would be stronger with more diverse languages.

2. The impact of auxiliary losses, such as weighting in multi-task setups (speech-to-text, text-to-speech), is not deeply analyzed or justified.

3. Comparisons against Qwen2.5-omni and other multimodal LLMs would improve to assess the quality of this work.

4. While pseudocode is provided, it does not clarify how key hyperparameters (number of groups, alignment weight, temperature) are chosen. Details on how slot embeddings and groupwise codebook splits are implemented could be improved.

**Questions:**

1. When dealing with multiple languages, the text frequency for individual languages might vary significantly. In such scenario, how to select an appropriate downsampling rate?

2. For text-to-speech experiment, while quantitative WER results are nice to see, how does the model perform on qualitative TTS metrics such as MOS, etc.?

3. When trained on Librispeech and evaluated on out-of-domain datasets such as commonvoice, etc, how does the WER evolve across aggressive downsampling?

4. Ablation on representation alignment is not very clear? What if the alignment loss is applied at more than one layers vs all layers vs at a fixed layer? Also, how important is the alignment loss starting from the speech-to-text task?

5. Again a broader question. How well does the proposed method generalize to out-of-domain benchmarks? In Section 5.3, the speech tokens from the ASR models are re-used, would these generalize to another domain such as commonvoice, etc.?

6. Appendix can improve by providing pseudocode that is easy to follow, highlighting the key innovations. What are the failure modes of this approach?

---

### Official Review · Reviewer_VMiM · 2025-11-01

**Soundness:** 2
**Presentation:** 2
**Contribution:** 2
**Rating:** 4
**Confidence:** 4

**Summary:**

This paper focuses on addressing the challenge of transferring knowledge from frozen text large language models (LLMs) to end-to-end pure-speech-token dialogue systems, with a core focus on optimizing the alignment between speech tokens and text tokens. It first identifies the problem of sequence length mismatch between speech and text tokens and questions whether averaging-based length matching is optimal. To solve the information bottleneck caused by aggressive downsampling when reducing speech token frame rate, the paper proposes an architecture that scales vector quantization (VQ) codebook capacity. Additionally, to mitigate the semantic gap between speech and text latent subspaces in LLMs, a representation alignment objective is introduced. Results show that the proposed method enables frozen text LLMs to perform pure speech-to-speech QA.

**Strengths:**

1. The research focuses on a critical issue of alignment between speech and text tokens in end-to-end speech dialogue models, which is essential for efficient knowledge transfer from text LLMs to speech modality.

**Weaknesses:**

1. Lack of novelty and incomplete literature review
* This paper proposes using a frozen LLM to reduce catastrophic forgetting of the LLM and transferring text knowledge to the speech modality through cross-modal alignment. However, this idea has already been put forward in numerous prior works, such as BLSP and AudioChatLLaMA, and the paper fails to discuss these relevant studies.
* The paper presents a "scalable VQ codebook," but the concept of grouped quantization was already proposed in VQ-VAE.
* The paper adopts InfoNCE loss to align speech and text at the representation level, yet this method was previously introduced in DIVA, and the paper also lacks a discussion comparing with this work.


2. Insufficient experimental details and unreliable conclusions
* For the Audio LM head—a core module in the proposed architecture—there is no description of its network structure or parameters, making the method difficult to reproduce.
* The paper lacks an introduction to evaluation metrics; for instance, it does not explain how "QA scores on Llama Questions" are calculated, nor does it cite relevant literature to support the metric’s validity.
* There are no case studies or subjective evaluation scores (e.g., Mean Opinion Score) for the quality of generated speech, making it impossible to assess whether the generated speech is practically usable.


3. Using discrete tokens as input Is not well-motivated and reduces practical applicability
* Existing speech-language models typically use continuous representations as input, but this paper provides no justification for the necessity of using discrete tokens. This makes the proposed method inconsistent with mainstream approaches.
* When calculating the average number of tokens per second, the paper uses the LibriSpeech dataset—a corpus of audiobooks characterized by clean audio and slow speaking rates. It does not verify whether the conclusions drawn from this dataset are applicable to conversational scenarios.
* The paper only validates discrete tokens on the ASR task. However, paralinguistic information (e.g., emotion, speaker traits) and non-speech sounds (e.g., music, background noise) are lost during the discretization process, which further limits the method’s generalizability.


4. Lack of discussion on limitations
* The paper insists on freezing the text LLM but does not discuss whether this choice imposes limitations on performance (e.g., whether fine-tuning part of the LLM backbone could alleviate the cross-modal semantic gap and improve results).
* In the speech→text task (Section 5.2), the paper observes a U-shaped WER curve (with optimal performance at 12.5–4.17 Hz) but only attributes poor performance at extreme frame rates to "the LLM struggling with redundant temporal resolution" (for high frame rates) or "excessively high information density per token" (for low frame rates). It fails to provide a deeper mechanistic analysis (e.g., how the LLM’s attention mechanism or positional encoding interacts with speech token sequence length to cause this phenomenon).

**Questions:**

see weaknesses

---

### Official Review · Reviewer_NdSm · 2025-11-01

**Soundness:** 2
**Presentation:** 2
**Contribution:** 2
**Rating:** 4
**Confidence:** 4

**Summary:**

This paper presents a simple yet effective approach for adapting a frozen text-based LLM to speech-to-speech QA. The authors investigate how the frame rate of speech tokens influences both speech understanding and generation, and propose using groups of FSQ to expand the effective codebook to nearly 300 bits per frame, thereby alleviating the information bottleneck at low frame rates. The paper further examines the optimal layer for representation alignment between speech and text modalities. Experimental results show that, with an appropriately chosen frame rate of 4.17 Hz, the adapted model—without any fine-tuning of the underlying text LLM—achieves strong question-answering performance directly in the speech-to-speech setting.

**Strengths:**

- **Effective Mitigation of Information Bottleneck:**

    The proposed strategy of splitting the codebook into n groups effectively alleviates the information bottleneck that arises at low frame rates, enabling richer speech representation.

- **Well-Justified Optimal Frame Rate:**

    The discovered optimal frame rate 4.17Hz has a nice explanation in Section 5.5 as it is slightly higher than average text token rate of 3.32Hz, covering the 0.8 and 2.2 range of text token rate.

**Weaknesses:**

- **Ambiguity in Section 5.3 Writing:**

    The first two paragraphs of Section 5.3 are somewhat confusing. The term *ASR* in the second paragraph actually refers to the **evaluation setup** rather than the **ASR stage** itself, which should be made explicit. Clarifying this distinction would improve readability and prevent misinterpretation.

- **Layerwise Distillation is Old Idea:**

    The proposed *layerwise distillation* analysis revisits an idea that has been explored in previous work, including studies on speech translation such as [1, 2]. The authors should better contextualize how their use of layerwise comparison differs from or extends these prior approaches.

- **Contradictory Results at Low Frame Rate:**

    Both the ASR and TTS components perform well at the downsample rate 1 (ds1), yet the speech QA results at the same rate are unexpectedly poor (Figure 6). This inconsistency needs an explanation.

- **Unfair Baseline Comparison in Section 5.8:**

    The comparison presented in Section 5.8 appears unfair, as the baseline systems are built upon **different base LLMs**. This makes it difficult to isolate the contribution of the proposed method from variations in underlying model capability.


[1] CKDST: Comprehensively and Effectively Distill Knowledge from Machine Translation to End-to-End Speech Translation (Lei et al., Findings 2023)

[2] Shimizu, S., Chu, C., Li, S., & Kurohashi, S. (2022). Cross-Lingual Transfer Learning for End-to-End Speech Translation. *Journal of Natural Language Processing*.

**Questions:**

- **Comparison to Cascaded Pipeline:**

    Since the text LLM remains frozen, why not adopt a cascaded approach (ASR → LLM → TTS) instead of direct speech-to-speech modeling?

- **Adaptation of Token2Wav:**

    How is the **token2wav** module from *CosyVoice2* adapted to accommodate your speech tokens?

- **Interpolation of Speech Tokens:**

    Have you experimented with **interpolating speech tokens** at lower frame rates, analogous to interpolation methods used for text tokens?

---

### Official Review · Reviewer_N8yM · 2025-11-01

**Soundness:** 2
**Presentation:** 1
**Contribution:** 1
**Rating:** 2
**Confidence:** 4

**Summary:**

This paper presents a method to address alignment between speech and text tokens to enable more effective transfer from a strong pretrained frozen text LLM to speech tasks such as dialogue.
The two key challenges here are length mismatch and representational alignment.
To address length mismatch, as they find that downsampling speech to match text resolution creates an information bottleneck, they instead propose scalable vector quantization (up to just under 300 bpf) with an efficient factorized audio LM head.
To align potentially distinct text and speech representational subspaces, they propose a contrastive alignment objective on intermediate LLM hidden states to pull speech and text representations into a shared latent space.
While experiments show improvements on downstream QA tasks compared to past models on those tasks, the proposed approaches are not ablated or compared to any others or past work on speech-text representation alignment, though both the length mismatch and aligning latent subspaces have previously been studied.

**Strengths:**

The motivation - aligning speech and text representations for length and subspace alignment - particularly for lightweight adaptation of pretrained speech or text models - is an important open challenge.

The model achieves competitive speech QA results with significantly fewer parameters and training data than past work.

**Weaknesses:**

- Insufficient discussion of and comparison to other methods for the two key challenges addressed here. See for example [Chen et al, Interspeech 2022](https://arxiv.org/abs/2204.03409) or [Le et al, ICML 2023](https://arxiv.org/abs/2301.11716) for each respectively, among others

- Only speech features from whisper-large-v3, which is heavily supervised for ASR potentially meaning these features are already somewhat aligned with text semantics, are used. Similarly, only text features from frozen Qwen3-4B are tried. It is unclear how much of the model's success comes from the use of this ASR-biased encoder, or the particular text LLM backbone, versus the paper's proposed techniques.

- The model is presented as a "pure-speech-token dialogue model", but the evaluation lacks any assessment of the generated audio quality. While the text-to-speech stage (Figure 5) and speech-to-speech stage (Figure 6) are evaluated on ASR-based content metrics (WER, QA), there are no subjective or objectives metrics on the synthesized waveforms and the claim of accurate synthesis is only supported anecdotally. Further, though motivated by dialogue, only single turn QA tasks are evaluated, making this somewhat misleading

**Questions:**

How dependent is this framework on the ASR-supervised encoder, and particular text LLM? The study should be replicated using a non-ASR-supervised encoder, such as a self-supervised model (e.g., HuBERT) or a standard codec encoder, and another text LLM, to provide additional exploration of the proposed alignment methods.

Were any other represenational alignment methods considered or explored? Particularly once length aligned, a variety of distance metrics could be used.

Given that this is a generative speech model, which  the authors use as the reason to "not compare against models that first generate text tokens as guidance before producing speech tokens", evaluation should likely include explicit evaluation of the generated speech quality through e.g. listening tests with MOS to confirm whether the proposed method affects synthesis quality.

---

### Official Review · Reviewer_AnoP · 2025-11-01

**Soundness:** 2
**Presentation:** 2
**Contribution:** 2
**Rating:** 2
**Confidence:** 5

**Summary:**

The paper investigates how speech token frame rates and representation alignment influence a frozen text LLM’s ability to process spoken inputs. It introduces a tokenizer architecture based on factorized scalar quantization (FSQ) with a parallel non-autoregressive audio head for efficient codebook scaling, and applies a contrastive representation alignment loss between speech and text embeddings at specific LLM layers. Through systematic experiments varying frame rates and alignment layers, the authors find that mid-range frame rates (around 4-6 Hz) yield optimal speech understanding performance. The paper argues that aligning speech and text through frame-rate matching and cross-modal representation alignment enables frozen text LLMs to perform end-to-end speech QA.

**Strengths:**

1. The paper's exploration of  “efficient transfer of text-based reasoning and knowledge into the speech domain” is timely and relevant for ongoing research in speech-language modeling.
2. The scalable tokenizer design combining factorized quantization and a lightweight non-autoregressive audio head is a reasonable improvement for handling large vocabularies at low frame rates.
3. Achieving pure speech-token QA using a frozen LLM and ≈2.5k hours of speech (with only ∼100-150M trainable parameters) is a strong result that suggests alignment-focused training can be data-efficient.

**Weaknesses:**

1. The paper’s hypothesis that “if we design a speech tokenizer whose frame rate closely matches the text token rate, would this lead to more effective alignment?” lacks strong theoretical grounding. The rate or count of text tokens does not correspond reliably to phonetic or temporal duration (for example, “elephant” and “cat” may each map to a single text token but differ substantially in acoustic length and frame count.) Consequently, the assumption that matching the frame rate of speech tokens to the rate of text tokens can meaningfully improve speech tokenization is not well supported by empirical evidence presented in the paper.

2. Prior work has explored bringing speech and text embeddings into a shared latent space, such as DM-Codec (Ahasan et al., 2024). The paper's representation alignment objective is similar to that of DM-Codec, that is, extracting hidden states from both modalities at selected layers, averaging over the temporal dimension, and obtaining normalized vectors. The most notable difference lies in using a contrastive objective instead of a cosine similarity-based objective (as in DM-Codec). However, the paper does not mention how their work improves upon or differs in contribution.

3. The linear interpolation of text embeddings for temporal stretching is not well-motivated. The paper assumes “alignment between speech and text is monotonic, so linear interpolation is reasonable,” yet provides no empirical validation or evidence that this approximation captures meaningful cross-modal relationships.

4. The tokenizer architecture combining factorized quantization and a non-autoregressive head is described as new, but similar designs have appeared in prior codec models such as NaturalSpeech3's FACodec (Ju et al., 2024). The paper does not isolate which component contributes to observed improvements or clarify how the design differs from established techniques.

5. Several experiments read more like ablation studies rather than core contributions. For example, the “length alignment” and “which LLM layer to align” analyses closely mirror the ablation experiments presented in SpeechTokenizer (Table 3: Results on SLMTokBench) and DM-Codec (D.4 Ablation Study: Impact of Different Distillation Layer). Framing these as contributions without introducing new insights beyond prior findings feels misplaced.

6. The “Comparison with Other Methods” section is limited to two baselines and omits stronger, more recent models. The paper should compare against established speech tokenizers and codecs (e.g., EnCodec, SpeechTokenizer, DAC, WavTokenizer, BigCodec) and speech-LLM approaches (e.g., CosyVoice2, Moshi, SLAM-Omni). Without these, it is difficult to evaluate the practical advantage of the proposed alignment and downsampling scheme.

**Questions:**

1. Please refer to the Weakness section for areas where further analysis or clarification would strengthen the paper.

2. Can the authors clarify how matching average text-token and speech-frame lengths theoretically leads to better alignment? Have they analyzed alignment quality beyond performance?

3. What are the concrete differences between the proposed representation alignment and prior objectives (e.g., DM-Codec's cosine alignment)? Can the authors provide clarification or comparison?

4. Can the authors provide controlled comparisons with recent strong baselines (EnCodec, DAC, WavTokenizer, BigCodec, SpeechTokenizer) under matched bitrates and codebook configurations?

5. How does the model perform under noisy or multilingual conditions? A small evaluation on non-English or real-world conversational speech would make the generality claim more convincing.

---

### Meta-Review · Area_Chair_nBNs · 2026-01-03

**Summary:**

The reviews were concerned with the following:
- unsubstantiated reasoning for design choices. E.g. the use of linear interpolation to approximately align the tokens between speech and text modalities
- unfair comparisons with inconsistent base-LLMs
- insufficient literature review to address similar works

**Reviewer Concerns:**

Authors did not provide a response

**Reviewer Scores:**

Not applicable, as authors did not respond.

---

### Decision · Program_Chairs · 2026-01-26

Reject